# Prosodic Differences in Women with the *FMR1* Premutation: Subtle Expression of Autism-Related Phenotypes Through Speech

**DOI:** 10.3390/ijms26062481

**Published:** 2025-03-11

**Authors:** Joseph C. Y. Lau, Janna Guilfoyle, Stephanie Crawford, Grace Johnson, Emily Landau, Jiayin Xing, Mitra Kumareswaran, Sarah Ethridge, Maureen Butler, Lindsay Goldman, Gary E. Martin, Lili Zhou, Jennifer Krizman, Trent Nicol, Nina Kraus, Elizabeth Berry-Kravis, Molly Losh

**Affiliations:** 1Roxelyn and Richard Pepper Department of Communication Sciences and Disorders, Northwestern University, Evanston, IL 60208, USA; josephcylau@northwestern.edu (J.C.Y.L.); janna.guilfoyle@northwestern.edu (J.G.); stephanie.crawford@northwestern.edu (S.C.); gracejohnson2024@u.northwestern.edu (G.J.); emily.landau@northwestern.edu (E.L.); jiayinxing2018@northwestern.edu (J.X.); mitra.kumareswaran@northwestern.edu (M.K.); sarah.ethridge@northwestern.edu (S.E.); maureen.butler@northwestern.edu (M.B.); lindsay.goldman@northwestern.edu (L.G.); j-krizman@northwestern.edu (J.K.); tgn@northwestern.edu (T.N.); nkraus@northwestern.edu (N.K.); 2Department of Communication Sciences and Disorders, St. John’s University, Queens, NY 11439, USA; marting@stjohns.edu; 3Department of Pediatrics, Neurological Sciences, and Biochemistry, Rush University Medical Center, Chicago, IL 60612, USA; lili_zhou@rush.edu (L.Z.); elizabeth_berry-kravis@rush.edu (E.B.-K.)

**Keywords:** *FMR1* premutation, autism, speech prosody

## Abstract

Evidence suggests that carriers of *FMR1* mutations (e.g., fragile X syndrome and the *FMR1* premutation) may demonstrate specific phenotypic patterns shared with autism (AU), particularly in the domain of pragmatic language, which involves the use of language in social contexts. Such evidence may implicate *FMR1*, a high-confidence gene associated with AU, in components of the AU phenotype. Prosody (i.e., using intonation and rhythm in speech to express meaning) is a pragmatic feature widely impacted in AU. Prosodic differences have also been observed in unaffected relatives of autistic individuals and in those with fragile X syndrome, although prosody has not been extensively studied among *FMR1* premutation carriers. This study investigated how *FMR1* variability may specifically influence prosody by examining the prosodic characteristics and related neural processing of prosodic features in women carrying the *FMR1* premutation (PM). In Study 1, acoustic measures of prosody (i.e., in intonation and rhythm) were examined in speech samples elicited from a semi-structured narrative task. Study 2 examined the neural frequency following response (FFR) as an index of speech prosodic processing. Findings revealed differences in the production of intonation and rhythm in PM carriers relative to controls, with patterns that parallel differences identified in parents of autistic individuals. No differences in neural processing of prosodic cues were found. Post hoc analyses further revealed associations between speech rhythm and *FMR1* variation (number of CGG repeats) among PM carriers. Together, the results suggest that *FMR1* may play a role in speech prosodic phenotypes, at least in speech production, contributing to a deeper understanding of AU-related speech and language phenotypes among *FMR1* mutation carriers.

## 1. Introduction

Prosody involves the use of intonation and rhythm to encode not only lexico-syntactic information (e.g., lexical stress and sentence structure), but also, importantly, to represent paralinguistic information such as speaker intent (e.g., sarcasm), emotion (e.g., anger), and emphasis (e.g., word focus in a sentence)—which are essential components of pragmatic (i.e., social) language [1,2]. Autism (AU) is a genetically based neurodevelopmental disorder that significantly affects social use of language, or pragmatics [3,4,5,6,7]. Among the areas of pragmatics impacted in AU is prosody [7].

Differences in prosody production in AU have been widely reported both in clinical reports (see [8] for a review) and acoustic characterizations (see [9] for a review). In particular, acoustic studies have found that autistic individuals typically exhibit higher and more variable pitch, less distinction in duration between stress and unstressed syllables, a slower speech rate, as well as greater intensity and longer phrase durations during speech production [8,10,11,12,13,14,15,16,17,18]. Recent correlational studies suggest that these prosodic characteristics are important markers of pragmatic difficulties in AU [19,20]. Perceptually, difficulties in prosodic processing are also evident in AU [17,21,22], which correlate with known disruptions in underlying auditory neurophysiological mechanisms implicated in AU [23,24]. In particular, a growing body of literature examining the neural frequency following response (FFR), a neurophysiological marker that comprehensively indexes the fidelity of neural auditory processing [25,26], has revealed inefficient neural tracking of pitch and temporal components of speech stimuli, as well as less stable responses to speech sounds in autistic individuals [23,24,27,28,29]. Along with the vast heterogeneity across individuals both in quality (e.g., affected prosodic dimensions) and severity of atypical prosody (see [10] for a review), results highlight a need for further investigation into the mechanistic pathways contributing to variability in prosody, both in terms of production and processing.

To understand the etiology and biological origins of prosodic and broader pragmatic differences in AU, first-degree relatives, particularly parents of autistic individuals, provide an apt model to examine the distilled expression of genetically meaningful traits associated with AU [19,30,31,32]. Strong evidence suggests that first-degree relatives of autistic individuals display elevated rates of subclinical traits that mirror AU features qualitatively, including pragmatic difficulties, collectively termed the broad autism phenotype (BAP) [31,32,33]. The abundance of evidence on subclinical differences in pragmatic abilities among clinically unaffected first-degree relatives indicate that AU-related genetic influences are expressed in this important clinical domain [33,34]. Specifically, for prosody, features that are qualitatively similar to those observed in autistic individuals (i.e., wider pitch range, slower speech rate, and a reduced tendency to adjust prosody to conversational partners) have been identified in parents of autistic individuals [19,35]. These findings implicate that prosodic differences may serve as an etiologically significant phenotype that may provide clues to the biological and genetic basis of AU traits. Furthermore, recent work suggests that speech processing, as measured by the FFR, could be a heritable biomarker of atypical prosody. Increased response latencies in parents of autistic children, strong parent–child associations in FFR responses, and associations with atypical prosody in both AU and the BAP support this potential biomarker [24].

A complimentary approach to studying the etiology and biological origins of AU-related traits involves comparisons with known genetic syndromes and more highly penetrant single-gene mutations (e.g., *CHD8*, *SCN2A*) associated with AU [36,37,38,39]. In particular, mutations in the Fragile X messenger ribonucleoprotein 1 (*FMR1*) gene, specifically expansions of the cytosine-guanine-guanine [CGG] trinucleotide repeat in the 5′ untranslated region (5′ UTR) of the *FMR1* gene located at Xq27.3, and the resulting disruptions to Fragile X messenger ribonucleoprotein (FMRP), constitute a highly important risk factor for AU symptomatology [40,41,42,43]. A full mutation of *FMR1* (>200 CGG repeats) results in transcriptional silencing of *FMR1* and loss or reduction in FMRP expression, which results in fragile X syndrome (FXS). Notably, FXS is the most common monogenic cause of AU [44,45,46]. Therefore, *FMR1* mutations and their impact on AU-related traits, including underlying neurobiological mechanisms, may offer valuable insights into the role of *FMR1* in AU. Indeed, FXS has been widely studied as a promising single-gene model for examining the gene–brain–behavior pathways associated with AU symptomatology [36,38,44,45], particularly in the domain of pragmatic language profiles, where significant phenotypic overlap has been documented among AU and FXS [47]. Furthermore, overlapping challenges across FXS and AU in pragmatic skills, such as underpinning cognitive mechanisms (e.g., social cognition, visual attention), prosodic speech features (e.g., atypical pitch, speech rate, rhythm), and greater rates of atypical speech disfluencies [48,49,50,51], provide further evidence of the role of *FMR1* in these pragmatic features [50,52,53].

While FXS is a rare condition (~1 in 4000 males and ~1 in 8000 females), the *FMR1* premutation (PM; 55–200 CGG repeats) occurs in roughly 1 in 150–250 women [54,55]. Women with a *FMR1* PM thus offer a robust model to study phenotypic profiles (including subtle AU-related traits) potentially linked to variations in the *FMR1* gene, without confounding factors of intellectual disability, developmental delays, and comorbidities common in FXS [56,57]. Importantly, pragmatic differences have been consistently documented among *FMR1* PM carriers [58,59,60]. Observational evidence specifically suggests that PM carriers make qualitatively similar pragmatic violations at the same rate as parents of autistic individuals, such as dominating or withdrawn conversational styles [34]. Further evidence suggests overlapping mechanistic processes affecting pragmatic profiles in both PM carriers and parents of autistic individuals, including differences in visual attention [61,62], social physiological arousal [63], and social cognitive abilities [47]. The pragmatic profiles related to AU in PM carriers have also been linked to increased AU symptom expression in their children with FXS [34], as well as variations in *FMR1*-related molecular genetics [64,65]. While these findings suggest that *FMR1*-related variation may play a role in AU-related pragmatic phenotypes observed in PM carriers, the relationship between prosodic properties observed in AU and the BAP during speech production, and their connection to *FMR1* variation, remains unclear. Similarly, neural perceptual processes related to prosody, and implicated in AU (i.e., the FFR), have yet to be examined. Given that FMRP is highly expressed in the auditory midbrain, the chief generator of the FFR response [66,67], there is a strong theoretical and empirical basis to suggest that atypical neural auditory processing may contribute to shared prosodic profiles observed in individuals with AU, the BAP, and *FMR1* conditions.

To disentangle the extent to which such speech factors may be uniquely associated with different mechanistic or genetic factors related to *FMR1* variation, this paper reports two studies aimed at characterizing the prosodic behavior of women with *FMR1* PM in both speech production and neural perceptual domains, with two distinct participant samples.

### 1.1. Study 1: Characterizing Prosodic Production in FMR1 PM Carriers

Study 1 examined the prosodic properties of speech among women with *FMR1* PM during a semi-structured narrative task (see Section 4.1.2 *Narrative Elicitation*) and compared their prosodic properties to those of controls and mothers of autistic individuals in a case–control design. Participants included 52 female *FMR1* PM carrier mothers (PM group), 98 mothers of autistic individuals (AU parent group), and 53 control mothers (control parent group) (Table 1, also see Section 4.1.1 *Study 1 Participants*). The groups were matched on age, verbal IQ, and the amount of language samples elicited (*p*s > 0.05, one-way ANOVA).

Focusing on measures of intonation and rhythm, prosodic domains impacted in AU [68] and where differences have been observed in in mothers of autistic individuals [19], we examined three questions:Are prosodic differences evident in *FMR1* PM?Are prosodic characteristics in *FMR1* PM similar to prosodic features associated with AU and the BAP?Are prosodic properties of speech among *FMR1* PM carriers associated with variability of the *FMR1* gene?

We conducted acoustic analyses (see Section 4.1.3 *Acoustic Analysis*) focusing on two core domains of speech prosody: speech intonation (i.e., vocal pitch variations across time) and speech rhythm (i.e., temporal regularities in the envelope of vocalic energy). Speech intonation analysis examined the fundamental frequency (F0), which is an acoustic correlate of pitch patterns, specifically its mean (*mean F0*) and range (*F0 range*). Speech rhythm analysis utilized an empirical mode decomposition (EMD) approach to quantify temporal regularities associated with rhythmic properties from the temporal envelope of the speech signal [69]. We focused on two acoustic variables yielded from the EMD analysis: *instantaneous frequency (ω1)* and *variance of the instantaneous frequency (var.ω1).* These variables index speech rate (i.e., how fast a person talks overall) and its variability (i.e., how talking speed varies within an utterance), respectively. Linear mixed-effect models were used to test whether each acoustic variable varied as a function of group.

Post hoc exploratory correlational analyses were performed to examine potential links between AU-related prosodic differences and *FMR1*-related variation (i.e., activation ratio, number of CGG repeats, and level of quantitative FMRP; see also Section 4.1.4 *Molecular-Genetic Analysis*). To maintain a conservative approach, we focused the correlational analyses only on analyzing acoustic metrics demonstrating group differences.

### 1.2. Study 2: Studying Speech Prosodic Processing Through Neural Frequency Following Responses to Prosodic Patterns

Study 2 examined the neural encoding of the acoustic properties of speech among women with *FMR1* PM, as measured by the frequency following response (FFR; see Section 4.2.2 *Frequency Following Response Elicitation and Data Processing*), comparing them to controls and mothers of autistic individuals.

Participants included 17 female *FMR1* PM carrier mothers (PM group), 18 mothers of autistic individuals (AU parent group), and 13 control mothers (control parent group) (Table 2, also see Section 4.2.1 *Study 2 Participants*). The groups were matched on age and Full Scale IQ (FSIQ).

Focusing on measures of neural encoding of prosodic-related acoustic cues, where differences in mothers of autistic individuals have emerged in prior literature, we examined the following questions:Is reduced accuracy in neural encoding of prosodic speech cues evident in *FMR1* PM?Are patterns of neural encoding speech prosody in *FMR1* PM similar to those observed in AU and the BAP?

FFR analyses were conducted on two primary variables of interest: the stimulus-to-response correlation between the fundamental frequencies of the stimulus and the response (*F0-SRC*) and the signal-to-noise ratio (*SNR*), which index the accuracy of neural encoding of prosodic cues in speech and the overall magnitude of neural activation during auditory stimulation, respectively. Linear models were used to test whether each variable varied as a function of group.

## 2. Results

### 2.1. Study 1 Results

Study 1 assesses the production of prosody in *FMR1* PM carriers by examining the acoustic properties of speech samples elicited in a semi-structured narrative task, as compared to parents of autistic individuals and controls. Acoustic properties were correlated with *FMR1*-related molecular-genetic measures.

#### 2.1.1. Group Differences: Speech Intonation

Measures of mean F0 and F0 range of PM, AU parent, and control parent groups are presented in Figure 1.

A main effect of group was identified in the linear mixed-effect model for F0 range (F(1,2) = 6.653, *p* = 0.001). Post hoc Wald tests revealed that both PM (W = 2.444, *p* = 0.022, FDR-adjusted) and AU parent groups (W = 3.616, *p* < 0.001, FDR-adjusted) had a wider F0 range than control parents. The F0 range of the PM and AU parent groups were not different from each other (W = 0.844, *p* = 0.399, FDR-adjusted).

The linear mixed-effect model for mean F0 did not yield a main effect of group (F(1,2) = 1.726, *p* = 0.178).

#### 2.1.2. Group Differences: Speech Rhythm

EMD-based measures of *ω1* (syllabic oscillation, indexing speech rate) and *var.ω1* (variability of syllabic oscillation, indexing variability in speech rate) of PM, AU parent, and control parent groups are presented in Figure 1.

A marginal effect of group was identified in the linear mixed-effect model for *var.ω1* (F(1,2) = 2.512, *p* = 0.081). Post hoc Wald tests revealed that both PM (W = 2.006, *p* = 0.075, FDR-adjusted) and AU parent groups (W = 1.962, *p* = 0.075, FDR-adjusted) had a marginally lower *var.ω1* (i.e., less variable syllabic oscillation) than control parents. The *var.ω1* measure of the PM and AU parent groups were not different from each other (W = 0.326, *p* = 0.745, FDR-adjusted).

The linear mixed-effect model for *ω1* did not yield a main effect of group (F(1,2) = 0.805, *p* = 0.447).

#### 2.1.3. Correlation Between Speech Rhythm and *FMR1* Molecular-Genetic Variation

Given that marginal group differences were evident for F0 range and *var.ω1* measures only, we conduced post hoc, exploratory correlation analyses only focused on the extent to which these measures were associated with FMRP, the number of CGG repeats, and activation ratio in the *FMR1* gene. While acknowledging the risk of both Type I and Type II errors, here we interpreted marginal differences given the expectation of pragmatic-related differences that are often subtle, but meaningful in AU parents and PM carriers [19,60]. Correlation results are presented in Table 3.

Notably, among PM carriers, less variable syllabic oscillation (i.e., lower *var.ω1*) was associated with a higher number of CGG repeats (rho = −0.311, *p* = 0.043) (Figure 2).

### 2.2. Study 2 Results

Study 2 assesses the neural encoding of prosodic cues in *FMR1* PM carriers via the frequency following response, as compared to AU parents and controls.

#### 2.2.1. Group Differences: Stimulus-to-Response Correlation

FFR measures of stimulus-to-response correlation of fundamental frequency (*F0-SRC*) of PM, AU parent, and control parent groups are presented in Figure 3.

The overall model did not reveal a main effect of group on *F0-SRC* (F(2, 45) = 1.74, *p* = 0.19). Pairwise comparisons showed that the AU parent group had marginally lower *F0-SRC* (i.e., reduced accuracy in pitch tracking) than control parents (b = 0.22, t(45) = 1.79, *p* = 0.08). The *F0-SRC* measure of the PM and AU parent groups were not different from each other (b = 0.15, t(45) = 1.28, *p* = 0.207), nor was the PM group significantly different from the control group (b = 0.07, t(45) = 0.59, *p* = 0.56).

#### 2.2.2. Group Differences: Signal-to-Noise Ratio

FFR measures of signal-to-noise ratio (*SNR*) of PM, AU parent, and control parent groups are presented in Figure 3. The overall model did not reveal a main effect of group on *SNR* (F(2, 45) = 0.40, *p* = 0.67). Pairwise comparisons did not reveal any differences between groups (AU parent vs. controls, b = 0.06, t(45) = 0.88 *p* = 0.39; AU parent vs. PM, b = 0.02, t(45) = 0.25, *p* = 0.80; PM vs. controls, b = 0.05, t(45) = 0.64, *p* = 0.53).

## 3. Discussion

This study investigated prosodic characteristics among women carrying the *FMR1* PM, aiming to examine the extent to which AU-related prosodic differences may be evident among women with the PM and whether such differences correlated with *FMR1*-related molecular-genetic variation (Study 1). A follow-up investigation (Study 2) explored whether the neural processing of speech prosodic cues might reflect a shared mechanistic contributor to atypical prosody spanning AU, the BAP, and *FMR1* PM carriers.

Subtle differences in key acoustic measures of intonation and rhythm were identified in both PM carriers and parents of autistic individuals. Specifically, acoustic differences in F0 range (indexing speech intonation) and variability of speech rate (indexing speech rhythm) were identified in the PM group relative to controls, reflecting differences in both fundamental components of prosody in PM carriers. The PM and AU parent groups performed comparably across these measures. Post hoc analyses further revealed a negative correlation between speech rate and CGG repeat length in the PM group. In Study 2, significant differences did not emerge in the fidelity of neural speech prosodic processing. However, a step-wise pattern was observed, with parents of autistic individuals demonstrating the lowest *F0-SRC*, followed by PM carriers, and then parent controls. This finding trended toward significance between the parents of autistic individuals and control parents.

### 3.1. Prosodic Differences in Speech Associated with FMR1 (Pre-)Mutation

Among speech domains where acoustic differences emerged, disruptions in speech rhythm have been previously implicated in FXS [48,49,50,70,71,72]. In FXS, differences in speech production include higher rates of cluttering, which may be present in up to 50% of males with FXS [51], contributing to poorer intelligibility in connected speech [73]. Whereas individuals with FXS most notably are characterized by an increased or fluctuating speech rate [50,70,71,72], particularly among those also meeting criteria for AU [48,49], our PM carriers demonstrated a contrasting pattern, with less variable speech rhythm observed. However, given that cluttering and other speech differences in FXS were reported to be associated with poorer nonverbal cognition [51], the qualitatively distinct acoustic differences found in PM carriers likely reflect subclinical variability in the context of intact cognitive abilities, among many other differences (e.g., neural, molecular-genetic) between *FMR1* pre- and full-mutation carriers.

Indeed, subclinical difficulties with speech motor functions have been implicated in the PM, associated with subtle disruptions to sensory–motor coordination centered in the cerebellum [74,75,76]. Relatedly, motor–speech differences have been documented in a subset of PM carriers who develop Fragile X-associated tremor/ataxia syndrome (FXTAS), a motor-based neurodegenerative condition unique to PM carriers, where speech dysfluencies have been reported in case studies [77,78,79]. While our cohort of PM carriers do not have FXTAS, the rhythmic differences identified in this study may point to pre-clinical markers of FXTAS and/or a partially overlapping profile of qualitatively similar, but subclinical, speech–motor differences that do not reach the clinical threshold for speech deficits [80].

Given the complex interrelationship between speech production and pragmatics [81,82], specifically the role of prosody in conveying crucial linguistic and paralinguistic information [83], the differences in rhythm and intonation found in PM carriers may be closely related to variations in their higher-order pragmatic skills. Prior evidence has indicated challenges in pragmatic language in PM carriers [34], with associated differences in skills that contribute to pragmatics, such as visual processing, executive functioning, and social cognition [61]. Given that prosodic differences contribute to pragmatic difficulties (e.g., in AU, William’s syndrome, and Down syndrome [11]), the prosodic differences identified in PM carriers may similarly contribute to pragmatic differences in this group.

### 3.2. Overlap in Prosodic Characteristics Among FMR1 Premutation Carriers and Parents of Autistic Individuals

Importantly, the results revealed relatively strong overlap in prosodic patterns among PM carriers and parents of individuals with AU. Whereas both groups showed only subtle differences from controls, similar types of differences were noted in each group, mirroring those documented in prior research [19]. Specifically, these differences included variations in speech rhythm and a wider F0 range, which is indicative of a more “sing-songy” intonation. Such intonational and rhythmic differences qualitatively mirror the prosodic patterns repeatedly identified in AU, both in English [19] and other languages [68,84]. Given noted phenotypic overlap between PM carriers and parents of individuals with AU [85], particularly in pragmatics [58,61,62,64,65], our results are the first to objectively demonstrate through acoustic analysis that AU-related prosodic profiles are expressed across these populations and may be influenced by the *FMR1* gene. Considering the important role of prosody in conveying pragmatic information such as emotion and communicative intent, the observed differences in prosody potentially linked to variation of the *FMR1* gene may contribute mechanistically to disrupted pragmatic language skills in *FMR1*-related conditions; these include not only the BAP and *FMR1* PM investigated in this study, but also AU and FXS.

### 3.3. Association Between AU-Related Rhythmic Properties of Prosody and FMR1 Variability

Only one acoustic measure of speech rhythm (but not intonation) was significantly associated with *FMR1*. Although the uncorrected statistical significance must be interpreted with caution, the association between a less variable speech rate and a higher number of CGG repeats is nevertheless intriguing.

While *FMR1* variation likely affects AU-related phenotypes through multiple pathogenic mechanisms, this association suggests that aspects of speech rhythm may be directly influenced by subtle reductions in FMRP. These effects are most clearly observed among PM carriers who carry CGG repeat lengths in the upper ranges, nearing 200. This finding further supports the hypothesis of varying mechanistic pathways contributing to component aspects of atypical prosody (i.e., intonation relevant versus rate/rhythm). Considering the previously discussed speech–motor-based difficulties in FXS and FXTAS, this suggests that *FMR1*-related molecular-genetic variation may contribute to rhythmic differences in speech, likely mediated by subclinical variation in speech–motor control. Indeed, individuals who carry higher repeat lengths are at a greater risk for developing FXTAS and have higher rates of motor difficulties [75,86,87]. While not examined in this cohort, the role of motor–speech control in contributing to speech rhythm variability should be examined in future research on *FMR1* PM carriers to further elucidate this theorized mechanistic pathway.

In the AU literature, speech rhythm has consistently emerged as an area marked by differences, not only in English but also in typologically distinct languages and different cultures, suggesting a stronger biological influence on this aspect of speech related to the AU-related prosodic profile [68,84]. Specifically, our group previously found that speech rhythmic differences in AU were invariant across two languages, despite drastic cross-linguistic typological and prosodic differences [68]. In contrast, AU-related differences in intonation were language-specific [68]. These cross-linguistic rhythmic similarities in AU may reflect a core phenotypic expression of an AU-related genotype that is relatively invariant to environmental and experiential modulations. Rhythm has been conceptualized as a central component of the neurobiological processes subserving speech processing and speech production [88,89,90], underscoring the potential significance of this domain regarding the present findings and broader research in AU [89,90,91]. Future studies examining prosody in PM carriers would benefit from cross-linguistic studies to examine the relationship between *FMR1* variation and both clinical and subclinical variations in speech rhythm across languages, while considering the influence of environmental factors such as language and culture.

### 3.4. No Differences in Neural Processing of Speech Prosodic Cues in FMR1 PM

Prior studies have demonstrated the important role of FMRP in the neural auditory system [41,91], which supports auditory perceptual processes contributing to prosodic ability and its development [24,92,93]. In particular, FMRP is highly expressed in the auditory midbrain [94], a central hub of the subcortical auditory system which determines the fidelity of acoustic cues encoded for downstream cortical speech processing, sensory–motor integration, and the development of speech, language, and broader auditory skills [95]. In mouse models of FXS, it has been demonstrated that subcortical disruptions in early sound processing are impacted by FMRP [66,67].

Despite such a well-documented role of FMRP in the auditory midbrain [96], our study did not reveal any significant differences between *FMR1* PM carriers or parents of autistic individuals from controls. Given the small sample size in Study 2, which limited the power to detect subtle subclinical differences between groups, a marginal finding nevertheless emerged showing reduced accuracy in speech sound encoding in AU parents. This was observed in the context of a qualitative stepwise pattern, suggesting that prosodic processing may be less impacted in PM carriers compared to AU parent group.

Considering the results of Studies 1–2 holistically, the lack of FFR differences in the PM group suggests neural auditory perceptual processes may play less of a role in shaping the prosodic and broader pragmatic phenotypes in PM carriers than in AU and the BAP. While correlational analyses between acoustic and FFR data were not feasible due to the inclusion of two largely distinct cohorts of participants across Studies 1 and 2, our current results provide further rationale to examine the production–processing link in *FMR1* mutation conditions. This will help to determine the potential salience of the gene–brain–behavior mechanistic framework pertaining to *FMR1* and speech prosody. In particular, we hypothesize that intonation-based differences in prosody (i.e., F0) may be more directly tied to FFR responses due to the FFR’s sensitivity to pitch encoding, whereas speech rhythm may reflect motor-based mechanisms.

### 3.5. Limitations and Future Directions

One limitation of the current study is our exclusive focus on speech samples elicited from a semi-structured language discourse. Whereas the semi-structured storybook context holds the advantage of eliciting roughly comparable speech across participants (i.e., all participants are generating narratives from the same pictures), it is important to examine prosody in more ecologically valid discourse contexts that represent social communication in everyday life, such as during unstructured conversations. Future work may also be fruitful in examining prosody in the PM across additional discourse contexts, as well as the extent to which the manifestation of *FMR1*-related prosodic differences may be modulated by environmental factors, such as social-economic status and cultural differences.

Also, as a hypothesis-driven study based on established findings in AU and the BAP [19], this study focused on a targeted but limited set of acoustic measures that index prosodic domains most prominently impacted in AU and the BAP, namely intonation and rhythm [68]. Future comparative studies should investigate a more comprehensive set of speech acoustic domains in relation to *FMR1*, not only to identify overlaps in the speech properties between AU/BAP and *FMR1* mutation conditions, but to delineate aspects of prosody that are undoubtedly unique in each group, which may provide valuable insights into connections with *FMR1* variation. Similarly, speech processing was evaluated using two well-established metrics that do not comprehensively capture the full potential of the FFR response. Recent advances in FFR research suggest several novel approaches to indexing the FFR that may be more sensitive to disentangling component factors of sounding encoding (i.e., timing, fundamental frequency, and harmonics) and provide a more nuanced lens into dynamic elements of sound processing.

The current study provides intriguing evidence of a potential subclinical pattern of speech prosody in PM carriers, possibly related to speech motor or cognitive factors, which warrants future investigation. There are well-established procedures that can sensitively capture speech-articulation fluency, such as diadochokinetic speech tests, which reflect the neurological ability to coordinate vocal articulators. Such tools will be useful in characterizing the speech–motor profile among *FMR1* PM carriers without FXTAS, which may help determine if speech-articulation differences represent a risk factor or prodromal feature of FXTAS. Further elucidating the role of oromotor ability in the context of naturalistic speech rhythm will also shed light on important brain-behavior pathways implicated in AU-associated prosodic differences, which could potentially inform early speech and language interventions. One future direction would be to also examine speech prosody and its undergirding mechanistic contributors in males with the *FMR1* PM to examine the extent to which prosodic differences covary with more obvious differences in other phenotypic domains observed in males with the *FMR1* PM.

## 4. Materials and Methods

### 4.1. Study 1 Materials and Methods

#### 4.1.1. Study 1 Participants

Participants included 52 female *FMR1* PM carrier mothers (PM group), 98 mothers of an autistic individual (AU parent group), and 53 control mothers (control parent group). Table 1 summarizes participant characteristics. All participants were native English speakers. PM carriers were recruited through study advertisements distributed to fragile X clinics and advocacy organizations, participant registries, and word of mouth. All PM carriers were mothers of a child with FXS, and PM status was confirmed via medical records.

Mothers of autistic individuals and parent controls were recruited as part of larger family genetic studies of AU and the BAP, in which control participants were screened for personal or family history of AU or language disorders. Both control and AU families were screened for genetic disorders, namely FXS, tuberous sclerosis, neurofibromatosis, and Rett syndrome. Additionally, only participants having no reported history of brain injury, major psychiatric disorder, or known hearing impairment were included. Mothers of autistic individuals were screened and excluded if they reported a personal history of an AU diagnosis and were included if their child had a formal diagnosis of AU. AU diagnoses were confirmed using the Autism Diagnostic Observation Schedule-2 (ADOS-2 [97]) and the Autism Diagnostic Interview-Revised (ADI-R [98]). Likewise, control participants were excluded if their child met diagnostic cutoff for AU on the ADOS-2. All ADOS-2 and ADI-R assessments were conducted by or overseen by personnel trained to research reliability standards. All participants had verbal IQ > 80 as per the Wechsler Abbreviated Scale of Intelligence (WASI [99]).

#### 4.1.2. Narrative Elicitation

Narrative samples were elicited using the 24-page wordless picture book, *Frog, Where Are You?* [100]. The book presents a story about a boy and his dog, who are searching for the boy’s missing pet frog. This book has been used extensively in studies of narrative discourse in AU and other neurodevelopmental disabilities [19,101,102,103,104,105]. Participants were asked to narrate the story, with pages of the book presented one by one to participants on a computer monitor, while their narrations were audio recorded. All narratives were recorded using a Logitech USB Blue Snowball Microphone (988-000067, Newark, CA, USA), placed approximately 8 inches from the participant’s mouth during the recording.

The recordings were first transcribed and segmented using the Praat version 6.4 [106] software into individual utterances defined by natural pauses. Each utterance hence corresponded to a maximal intonational phrase (i.e., the highest unit of structure within linguistic models of the prosodic hierarchy). Utterances were excluded if they contained any of the following: character speech, a question, unfinished words, an interruption by the examiner, fewer than two words, were shorter than 1 s, unintelligible speech, speech directed towards someone else in the room and/or not related to the narrative, and abandoned utterances.

An average of 41.4 utterances per participant were elicited and included in the analysis, with a range of 15–120 across participants. The number of utterances included did not statistically differ across groups (one-way analysis of variance: F(2,200) = 1.68, *p* = 0.19). Nevertheless, the inter-participant variability in the number of utterances elicited was accounted for statistically by the inclusion of a by-subject random effect in the statistical model (detailed in Section 4.1.5 *Study 1 Statistical Analysis*).

#### 4.1.3. Acoustic Analysis

Speech intonation analysis focused on the fundamental frequency (F0), which represents the frequency of vocal fold vibration that is an acoustic correlate to pitch. For each utterance, its F0 contour was derived using an autocorrelation-based procedure in Praat [106], with a pitch tracking range of 130–400 Hz appropriate for females [19]. Logarithmic transformation was applied to F0 values to approximate the scale on which pitch is perceived. After applying the logarithmic transformation, two F0-based measures were calculated: *1. Mean F0*, which measures the rate of vocal fold vibration and is the acoustic correlate of pitch, i.e., how “high” or “low” an individual’s voice is; *2. F0 Range*, calculated by subtracting the maximum and minimum F0 values obtained from each utterance, indexes the extent to which an individual’s pitch varies during speech.

Speech rhythm analysis focused on assessing two important metrics of temporal regularities in the envelope of vocalic energy, namely, speech rate (i.e., how fast a person talks), and its variability (i.e., how talking speed varies within an utterance). An empirical mode decomposition (EMD) approach was employed to represent temporal regularities correlating to rhythmic properties of the speech signal [69]. For each utterance, the vocalic energy amplitude envelope was first derived by chunking raw time series into consecutive bins of 1 s. The time series of each chunk was filtered with a passband of 400–4000 Hz to de-emphasize non-vocalic energy such as glottal energy and obstruent noise. The bandpass-filtered signal was then low-pass filtered with a cutoff of 10 Hz to represent the envelope. The intrinsic mode functions (IMFs) of the envelope were then derived using EMD, representing syllabic fluctuations relevant to speech rhythm [69]. We selected a bin duration of 1 s to maximally eliminate the representations of slower prosodic information (e.g., intonation) and mixtures of tempos and variations in rhythmicity not relevant to the syllabic rhythm [69]. We focused on: *3. the instantaneous frequency (ω1)* and *4. the variance of the instantaneous frequency (var.ω1)* of the first intrinsic mode function (IMF1), which represents oscillation in the envelope related to the rate of syllables as well as its variability (i.e., stability). Measures of *ω1* and *var.ω1* were derived using a Hilbert transform on the IMF1 as the time derivative of the instantaneous phase.

#### 4.1.4. Molecular-Genetic Analysis

Blood and buccal samples were obtained for DNA isolation, and DNA genotyping was performed at the Rush University Molecular Genetics Diagnostic Laboratory. *FMR1* molecular-genetic variation was characterized in 38 PM carriers, including CGG repeat length and activation ratio (i.e., the proportion of cells with the normal *FMR1* allele as the active allele). *FMR1* genotyping was performed using a triplet-primed Polymerase Chain Reaction (PCR) kit (Asuragen, Austin, TX, USA) to accurately determine CGG expansion repeat length [107]. Methylation status and activation ratio were determined using a methylation PCR kit (Asuragen, Austin, TX, USA) which distinguishes the active and inactive allele based on methylation pattern [108]. FMRP levels were assayed in lymphocytes isolated from blood using Luminex Technology [109] and quantified as FMRP concentration (i.e., quantitative FMRP, in ng/µL).

#### 4.1.5. Study 1 Statistical Analysis

Group differences in each acoustic variable (at the utterance level) were examined using a linear mixed-effect model (LMM), with group as a fixed factor. A by-participant random factor was included to account for the different number of utterances elicited from each participant. Since both intonation and rhythm vary as a function of utterance duration, the duration of each individual utterance was included in the LMM as a covariate. While the groups were marginally different in verbal IQ (*p* = 0.057, with post hoc tests revealing AU parents had marginally lower verbal IQ than controls; see also Table 1), the inclusion of verbal IQ in the LMM did not strengthen any of the models (*p*s > 0.467, likelihood ratio test), and was thus excluded.

A post hoc, exploratory analysis was performed to examine the extent to which acoustic variables demonstrating group differences were associated with molecular-genetic variables (quantitative FMRP, activation ratio, the number of CGG repeats). Since molecular-genetic data was only available for the PM group, correlation analysis was performed within the PM group only. Spearman’s correlation was used, considering that both linear and non-linear relationships between these molecular-genetic variables and phenotypic outcomes have been previously identified in the literature [63,110,111,112].

### 4.2. Study 2 Materials and Methods

#### 4.2.1. Study 2 Participants

Participants of Study 2 included 17 female *FMR1* PM carrier mothers (PM group), 18 mothers of an autistic individual (AU parent group), and 13 control mothers (control parent group). Table 2 summarizes the participant characteristics. Recruitment sources and procedures as well as exclusion and inclusion criteria were the same as those in Study 1. Only two participants overlap between Studies 1–2 (both from the PM group).

#### 4.2.2. Frequency Following Response Elicitation and Data Processing

Frequency following responses (FFRs) were elicited electrophysiologically using the Continuous Acquisition Module of the Intelligent Hearing Systems Duet. The FFR reflects the integrated activity of neurons of the cortical and subcortical auditory system as it encodes auditory signals, including essential properties of speech sounds such as onset, spectral amplitude, and pitch [113]. Here, electrophysiological responses were elicited from a naturally produced syllable /ja/. The syllable has a duration of 230 ms, with a resynthesized F0 contour that rises from 130 to 220 Hz, which reflects a typical intonation pattern of questions in English [24]. With Etymotic ER3C insert earphones (Etymotic Research), a total of 2400 sweeps of the syllable were presented to the participant’s right ear in alternating polarities, at a rate of 3.3 Hz and intensity of 70 db SPL. During electrophysiological testing, participants sat in a comfortable chair in a quiet room while watching a TV show or movie of their choice at a low volume, consistent with established FFR collection protocols [24,113,114].

Evoked electrophysiological responses were recorded from the scalp using a vertical electrode montage (Cz-ipsilateral earlobe, aFz ground) with Ag-AgCl electrodes. Contact impedances were maintained below 5 kOhms. Electrophysiological responses were digitized at 20,000 Hz and bandpass filtered online from 70 to 3000 Hz (12 dB/octave roll off). Offline, electrophysiological responses were epoched according to the onset of each syllable, encompassing pre- and post-stimulus windows of 50 ms each (i.e., 330 ms total). Epochs with activities of ±35 µV in amplitude were considered artifacts and discarded. The first 500 artifact-free epochs from each polarity (1000 trials total) were then averaged to represent the FFR.

#### 4.2.3. Frequency Following Response Data Analysis

We focused on two FFR metrics to assess neural encoding of prosodic-related acoustic cues, namely (1) stimulus-to-response correlation of the fundamental frequency (*F0-SRC*) and (2) signal-to-noise ratio (*SNR*).

We first assessed how faithfully the intonation patterns of the stimulus, which are crucial for conveying prosodic cues, are encoded by the brain by computing the stimulus-to-response correlation (SRC) between the F0 contour encoded in the FFR and the F0 contour of the stimulus [113]. A fast Fourier transform-based (FFT) procedure was used to estimate the F0 contour of the FFR and that of the stimulus, respectively, using 50 ms Hanning windows each overlapping by 49 ms. The *F0-SRC* is taken as the Pearson correlation coefficient (r) between the stimulus and response F0 contours (across all Hanning windows), which indicates the similarity between the stimulus and response F0 contours in strength and directionality.

To assess the overall magnitude of neural activation during auditory stimulation [113], we derived the SNR from each FFR. To derive the SNR, the root mean square (RMS) amplitudes of the FFR and the pre-stimulus baseline period of the waveform were first recorded. RMS amplitudes were the mean absolute values of all sample points of the waveform within the respective time windows, in µV. The *SNR* presents the quotient of the FFR RMS amplitude and the pre-stimulus RMS amplitude.

Computation of both *F0-SRC* and *SNR* was performed using customized MATLAB 2023a (The MathWorks, Inc., Natick, MA, USA) scripts following conventions established in the FFR literature [95,113].

#### 4.2.4. Study 2 Statistical Analysis

Group differences in each FFR variable (averaged over 1000 trials) were examined using a linear mixed-effect model (LMM), with group as a fixed factor. There were no significant differences between groups in age or IQ; thus, these variables were excluded as covariates.

## 5. Conclusions

This study’s findings enhance the current understanding of speech prosodic characteristics, not only specific to *FMR1* PM, but also in AU, where the *FMR1* gene has been implicated. By demonstrating differences in representative domains of prosody among PM carriers that overlap with AU-related prosodic profiles, findings provide insight into potential etiological pathways that may influence high-order social communication differences observed in *FMR1*-related conditions.

## Figures and Tables

**Figure 1 ijms-26-02481-f001:**
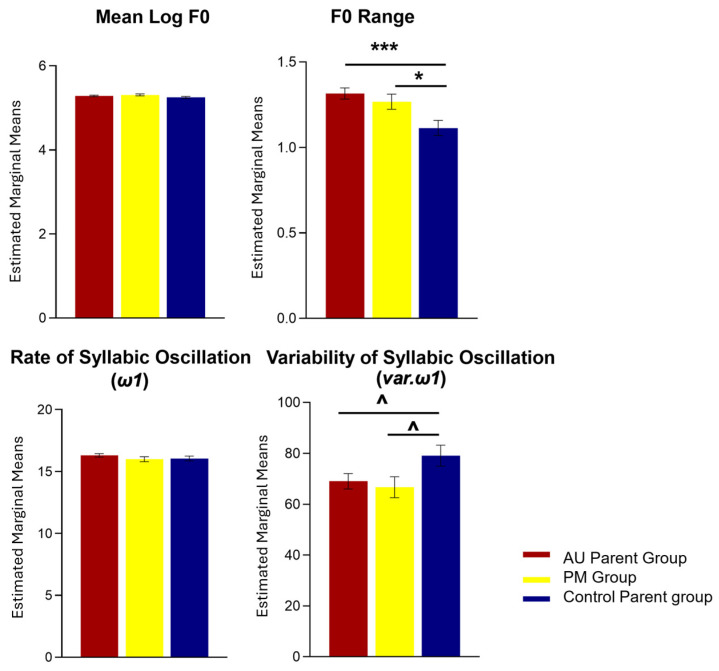
Estimated marginal means of acoustic measures indexing speech intonation (mean F0, F0 range) and rhythm (*ω1*—rate of syllabic oscillation; *var.ω1*—variability of syllabic oscillation) in PM, AU parent, and control parent groups. *** *p* < 0.001, * *p* < 0.05, ^ *p* = 0.1 in post hoc Wald tests, False Discovery Rate-corrected.

**Figure 2 ijms-26-02481-f002:**
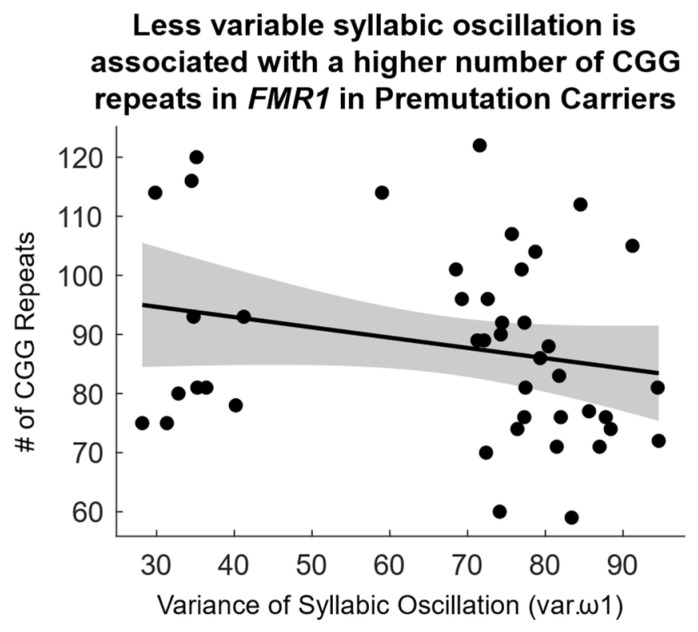
Association between variance of syllabic oscillation (*var.*ω1) and number of CGG repeats among *FMR1* PM carriers.

**Figure 3 ijms-26-02481-f003:**
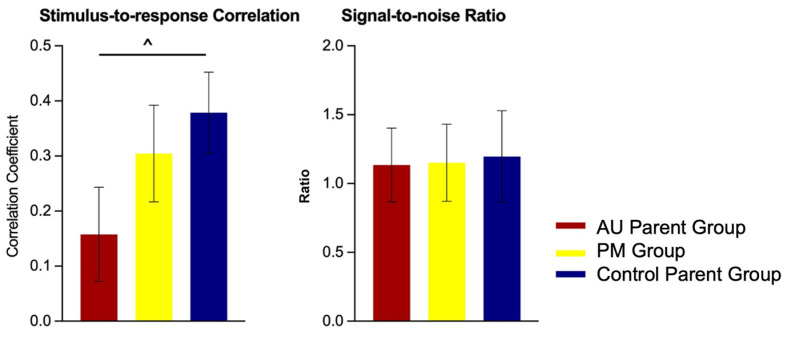
Estimated marginal means of FFR measures indexing the fidelity of neural processing of speech prosodic cues, stimulus-to-response correlation, and signal-to-noise ratio in PM, AU parent, and control parent groups. ^ 0.05 < *p* < 0.1.

**Table 1 ijms-26-02481-t001:** Demographic information Study 1. ^ 0.05 < *p* < 0.1, one-way analysis of variance.

Mean (SD)	AU Parent Group (*n* = 98)	PM Group (*n* = 52)	Control Parent Group (*n* = 53)
Chronological Age	46.43 (8.72)	46.40 (10.97)	44.00 (8.24)
Verbal IQ ^	110.99 (12.35)	113.71 (10.50)	115.48 (9.98)
Number of Utterances Elicited	43.78 (17.44)	38.87 (20.27)	39.60 (15.60)

**Table 2 ijms-26-02481-t002:** Demographic information Study 2.

Mean (SD)	AU Parent Group (*n* = 18)	PM Group (*n* = 17)	Control Parent Group (*n* = 13)
Chronological Age	52.14 (10.17)	52.67 (12.73)	47.72 (7.07)
FSIQ	115.11 (11.69)	115.29 (11.53)	117.31 (10.07)

**Table 3 ijms-26-02481-t003:** Correlations between prosodic features and *FMR1*-related variation. * *p* < 0.05, uncorrected.

Spearman’s ρ (*p*)	Quantitative FMRP	Activation Ratio	Number of CGG Repeats
Pitch (F0) range	0.142 (0.394)	−0.070 (0.668)	0.175 (0.261)
Syllabic Oscillation Variability (*var.ω1*)	−0.052 (0.755)	−0.102 (0.531)	−0.311 (* 0.043)

## Data Availability

Anonymized numeric data and all statistical codes will be made available at OSF upon publication of the paper.

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
