# Peer review of "Prosodic Differences in Women with the FMR1 Premutation: Subtle Expression of Autism-Related Phenotypes Through Speech"

_ijms, 2025, doi:10.3390/ijms26062481_

Round 1

Reviewer 1 Report (Previous Reviewer 3)

Comments and Suggestions for Authors

In this revision, the author made several changes and the quality of whole manuscript do improve a lot. The overall quality is suitable for potential publication. However, the discussion part is too reduntent, the author could consider precise it.

Author Response

In this revision, the author made several changes and the quality of whole manuscript do improve a lot. The overall quality is suitable for potential publication. However, the discussion part is too redundant, the author could consider precise it.

  • We appreciate this suggestion and have revised the Discussion to be more concise. Materials appearing in both in the Introduction and the Discussion have been trimmed as appropriate.

Reviewer 2 Report (Previous Reviewer 2)

Comments and Suggestions for Authors  

In general, I believe the introduction is too long. It should be kept in perspective that this is an original article, and it develops an introduction with 66 references, which could be excessive or suggest a need for a more direct development of ideas. I feel the introduction falls into basic explanations rather than developing the interrelationship between the variables of interest. For example:

• Lines 59-65: These are theoretical explanations of a variable.

• Line 110-111: This is a very basic explanation of a Mendelian inheritance pattern.

• Lines 130-148: How do the authors believe this information contributes to the clarity and support of their research problem?

• Lines 88-89: I suggest reviewing the locus of the FMR1 gene and improving the description. It seems a bit confusing to refer to a 5’ UTR position of a chromosome, as UTRs are more associated with the structure of a gene in terms of mRNA rather than genomic DNA.

• Line 169: What does “PS” mean?

• Table 2: Are the authors referring to general IQ or verbal IQ?

• Lines 426-439: The authors omit the use of already established abbreviations for premutation, autism, and broad autism phenotype. This can be observed in other parts of the text (e.g., line 474).

Line 518: What brand of microphone was used? What software was used for audio processing?

Line 566: A brief description of the method used to determine CGG repeats could be valuable to understand the validity of the quantification better.

In Study 1, there is a total of 203 participants, 52 from the PM group; however, only 38 from the PM group were analyzed. How do the authors justify that this characterization is sufficient for making comparisons attributable to the PM state, beyond the correlation analysis with CGG repeats?

In Study 2, it is not clarified whether genomic analysis was performed, nor how many participants were analyzed.

Other observations:

I recommend that the authors use People First Language. Instead of “autistic individuals,” say “persons with autism.” More at https://mdsc.org/programs/people-first-language/.

Comments on the Quality of English Language

I suggest that the English writing be reviewed in more detail, as some expressions are somewhat difficult to follow or could convey a more direct idea. For example:

• Line 35: The word “however” seems unnecessary.

• Lines 37-38: It is unclear whether this refers to differences or if it implies that the differences are the cause of a particular finding.

• Line 48: The sentence could be omitted, and only prosody could be mentioned as an example (such as prosody).

• Lines 50-51: It is unclear if “This” refers to the reference or the authors’ work.

• Lines 360-361: The expression seems repetitive: “…intonation in PM carriers… relate to differences in PM carriers…” It could be clearer: “Differences in rhythm and intonation found in PM carriers may be strongly related to differences in their higher-order pragmatic skills.”

• Lines 364-366: This could be more direct: “Given that prosodic differences have been shown to contribute to pragmatic difficulties (e.g., in AU, Williams syndrome, and Down syndrome), they may also contribute to such difficulties in PM carriers.”

• Line 478: A verb seems to be missing in reference to FXTAS.

Author Response

  1. In general, I believe the introduction is too long. It should be kept in perspective that this is an original article, and it develops an introduction with 66 references, which could be excessive or suggest a need for a more direct development of ideas. I feel the introduction falls into basic explanations rather than developing the interrelationship between the variables of interest. For example:
    • In this revision, the Introduction section has been significantly streamlined to improve the flow of the presentation of the formulation of our research question. The introduction is now 30 lines shorter than the previous version. We appreciate this suggestion from the reviewer.
  2. Lines 59-65: These are theoretical explanations of a variable.
    • Thank you for pointing this out. We have revised this paragraph to focus on prosodic processing in AU, rather than explaining what the FFR is. We have moved the explanations to section 4.2.2 of the Methods section.
  3. Line 110-111: This is a very basic explanation of a Mendelian inheritance pattern.
    • We have removed this sentence, given our interpretation to the reviewer’s suggestion that this information is not necessary for the target audience of this Journal.
  4. Lines 130-148: How do the authors believe this information contributes to the clarity and support of their research problem?
    • We agree with the reviewer that this information is not necessary in the Introduction. We have removed this paragraph in the revision and moved elements of the information to section 3.4 of the Discussion as appropriate.
  5. Lines 88-89: I suggest reviewing the locus of the FMR1 gene and improving the description. It seems a bit confusing to refer to a 5’ UTR position of a chromosome, as UTRs are more associated with the structure of a gene in terms of mRNA rather than genomic DNA.
    • Thank you for pointing out this error in the description of FMR1 mutations. We have revised the description to more accurately delineate the 5 ‘UTR position as the location of the CGG expansion on the FMR1 gene.
  6. Line 169: What does “PS” mean?
    • Thank you for spotting this error. The p (i.e., p-value) in “ps” is now italicized.
  7. Table 2: Are the authors referring to general IQ or verbal IQ?
    • We have clarified that we refer to a full-scale IQ score in study 2 analyses.
  8. Lines 426-439: The authors omit the use of already established abbreviations for premutation, autism, and broad autism phenotype. This can be observed in other parts of the text (e.g., line 474).
    • We regret this oversight and have replaced all instances of the fully spelled out forms with their established acronyms. Additionally, see response below re: autism terminology and our decision to utilize the abbreviation AU, rather than ASD.
  9. Line 518: What brand of microphone was used? What software was used for audio processing?
    • Clarified as Logitech USB Desktop Microphone (980186-0403), with transcriptions and segmentation performed in the Praat software.
  10. Line 566: A brief description of the method used to determine CGG repeats could be valuable to understand the validity of the quantification better.
    • Additional details were added to clarify the methods used for FMR1-molecular genetic analyses.
  11. In Study 1, there is a total of 203 participants, 52 from the PM group; however, only 38 from the PM group were analyzed. How do the authors justify that this characterization is sufficient for making comparisons attributable to the PM state, beyond the correlation analysis with CGG repeats?
    • As we pointed out in section 4.1.1, “all PM carriers were mothers of a child with FXS, and PM status was confirmed via medical records”. Therefore, as we now clarify in line 136, we present a case-control design, with which a comparison between PM and controls who do not have PM are sufficient to attribute differences to the PM state, even without a direct molecular genetic measurement. The post-hoc correlational analysis with CGG repeats of a subgroup of PM (38 out of 52) provides converging evidence to further support a link between prosodic differences with PM, but differences in PM identified stand on their own even without the post-hoc correlational findings given our case-control comparison.
  12. In Study 2, it is not clarified whether genomic analysis was performed, nor how many participants were analyzed.
    • No genomic analysis was performed in Study 2, and were hence not reported.
  13. I recommend that the authors use People First Language. Instead of “autistic individuals,” say “persons with autism.” More at https://mdsc.org/programs/people-first-language/.
    • In our work with stakeholders we have found that in recent times the majority of autistic individuals in the United States (where our participating families are from) prefer identify-first terminology, consistent with several recent studies (e.g., Taboas et al., 2023, Botha et al. 2023). While we recognize that person-first language is well intentioned, many autistic individuals find this terminology stigmatizing whereas they see identity-first terminology more reflective of their neurodiversity. In line with current practice in the autism field, we have therefore chosen to use identify first terminology. In this same vein, we have chosen to use AU to abbreviate autism, rather than ASD reflecting autism spectrum disorder, to honor stakeholder preferences and in support of a neurodiversity model of understanding autism (where the term “disorder” has been criticized).
  14. I suggest that the English writing be reviewed in more detail, as some expressions are somewhat difficult to follow or could convey a more direct idea.
    • We have performed thorough editing for language in this revision, where all reviewer’s suggestions below have been addressed.
  15. Line 35: The word “however” seems unnecessary.
    • Removed
  16. Lines 37-38: It is unclear whether this refers to differences or if it implies that the differences are the cause of a particular finding.
    • We have clarified this sentence as follows: “Together, results implicate a role of FMR1 in speech prosodic phenotypes, at least in its production, which helps to refine understanding of the expression of AU-related speech and language phenotypes among FMR1-mutation carriers.”
  17. Line 48: The sentence could be omitted, and only prosody could be mentioned as an example (such as prosody).
    • We feel this the sentence “Among the areas of pragmatics impacted in AU is prosody” is essential here because there are many other aspects of pragmatics impacted in AU, with prosody only one area our study focuses on.
  18. Lines 50-51: It is unclear if “This” refers to the reference or the authors’ work.
    • Replaced “this” with “Particularly, acoustic studies”
  19. Lines 360-361: The expression seems repetitive: “…intonation in PM carriers… relate to differences in PM carriers…” It could be clearer: “Differences in rhythm and intonation found in PM carriers may be strongly related to differences in their higher-order pragmatic skills.”
    • Thank you for the suggestion, it has been implemented.
  20. Lines 364-366: This could be more direct: “Given that prosodic differences have been shown to contribute to pragmatic difficulties (e.g., in AU, Williams syndrome, and Down syndrome), they may also contribute to such difficulties in PM carriers.”
    • Revised as “Given that prosodic differences contribute to pragmatic difficulties (e.g., in AU, William’s syndrome, and Down syndrome), prosodic differences identified in PM carriers may also contribute to pragmatic differences in this group.”
  21. Line 478: A verb seems to be missing in reference to FXTAS.
    • Thank you for spotting this clunky sentence. This sentence is revised as follows: “Such tools will be useful in characterizing the speech-motor profile among FMR1 PM carriers without FXTAS, which may help determine if speech-articulation differences represent a risk factor or prodromal feature of FXTAS.”

References:

Botha, M., Hanlon, J., & Williams, G. L. (2023). Does language matter? Identity-first versus person-first language use in autism research: A response to Vivanti. Journal of autism and developmental disorders, 53(2), 870-878.

Taboas, A., Doepke, K., & Zimmerman, C. (2023). Preferences for identity-first versus person-first language in a US sample of autism stakeholders. Autism, 27(2), 565-570.

Reviewer 3 Report (Previous Reviewer 1)

Comments and Suggestions for Authors

No further revisions

Author Response

Thank you for the review. We appreciate your insightful comments which we believe have vastly improved our manuscript.

Round 2

Reviewer 2 Report (Previous Reviewer 2)

Comments and Suggestions for Authors

Maybe it is a matter of the journal’s style, but there should be a space between the word and the bracket with citations.

I appreciate the effort you have made in addressing the comments and suggestions provided during the review process.

This manuscript is a resubmission of an earlier submission. The following is a list of the peer review reports and author responses from that submission.

Round 1

Reviewer 1 Report

Comments and Suggestions for Authors

The article explores a relevant and innovative topic, focusing on prosodic differences in women carrying the FMR1 premutation and their mechanistic implications for ASD-related phenotypes. The structure is well-organized, and the data are presented clearly, supported by appropriate methods.

Strengths are:

Originality: The topic is innovative and contributes to understanding ASD-related phenotypes concerning the FMR1 premutation.

Robust Methodology: The techniques used for prosodic and genetic analysis are current and well-described.

Significant Results: The findings provide valuable insights into associations between FMR1 variability and prosodic characteristics.

However there are some areas for improvement, as follow:

- Abstract

While informative, the abstract could be more concise. For example, some detailed methodological information can be omitted.

- The introduction provides comprehensive background but could be streamlined to avoid redundancy.

- The article provides useful details about the clinical characteristics of participants, but some aspects could be clarified or strengthened to address your concerns:

- Participants were excluded if they had a personal or family history of ASD, language disorders, or genetic conditions such as full Fragile X syndrome, tuberous sclerosis, neurofibromatosis, or Rett syndrome.

- Only individuals with a verbal IQ above 80, confirmed using the Wechsler Abbreviated Scale of Intelligence (WASI), were included.

- It is unclear whether standardized tools were used to evaluate the intensity and severity of ASD symptoms in participants.

- The protocol description does not explicitly mention the use of specific tests, such as the Autism Diagnostic Observation Schedule (ADOS), to rule out ASD diagnoses.

- About the ASD Screening:

The authors mention that participants and their families were evaluated for genetic and neurodevelopmental disorders, but it is not specified who performed these assessments (e.g., clinicians with ASD expertise) or what criteria were used.

- Include Detailed Assessments:

Specify whether standardized tools such as ADOS or Autism Diagnostic Interview-Revised (ADI-R) were used to rule out ASD in participants and their families. This would strengthen the validity of the findings and eliminate the risk of including participants with unrecognized subclinical symptoms.

- Describe the Evaluators:

Indicate whether assessments were conducted by clinicians with ASD expertise or professionals specifically trained in this area. This would enhance confidence in the screening's accuracy.

- Quantify Symptoms:

Using tools like ADOS could not only help exclude ASD diagnoses but also provide useful measures to quantify any subclinical traits, enriching the analysis.

- Discussion of Importance:

The discussion section could include comments on the importance of using standardized tools to confirm or exclude ASD and the implications of subclinical traits for the study's findings.

- Data Presentation

Some figures (e.g., Figure 1) lack a clear interpretation. Including a brief summary of what the data show would be helpful.

The data on prosodic rhythm variability could be better integrated into the discussion.

- While the discussion links the results to the study's objectives, further exploration of clinical implications and future directions would strengthen the manuscript.

- Some passages contain complex terminology that could be simplified for greater readability.

- Minor typographical errors are present (e.g., "intonational" instead of "intonation").

- Table 1: While it effectively summarizes demographic data, the statistical significance of the results could be emphasized further.

- Methods: Although detailed, some descriptions, such as those on utterance segmentation, could be condensed.

- Results: Providing more clarification on how the findings compare to previous studies could enhance understanding.

Reviewer 2 Report

Comments and Suggestions for Authors

A more detailed and organized description of each group's inclusion and exclusion criteria is suggested, as well as the tests and characterizations performed on the participants. It is also recommended that these results be presented in a table. This information will help assess the homogeneity of the compared groups, particularly regarding factors that could influence prosody, such as educational level, having children with autism, the language spoken and heard in their immediate context, mental health, and genetic characterization.

While the introduction is engaging and comprehensive, it may be considered too lengthy for the type of article presented. At times, it reads more like a review than a way to establish the principles underpinning the research problem. Explanations such as those in lines 61–69 seem to elaborate on assertions made in lines 55–60.

Lines 97–102 do not appear essential to understanding the problem, as it has already been mentioned that the gene of interest is located on the X chromosome. Similarly, in lines 103–115, revisit topics addressed in lines 60–69.

Reviewer 3 Report

Comments and Suggestions for Authors

The manuscript by Joseph C.Y. Lau and colleagues focuses on prosodic speech characteristics in women carrying the FMR1 PM, as well as parents of children with autism spectrum disorder. The study uses linear mixed-effect models and post-hoc Wald tests to examine group differences and explores correlations between speech features and FMR1-related genetic markers. They find that both PM and ASD Parent groups exhibit a wider F0 range than controls, and the PM group shows a marginally lower variability in syllabic oscillation compared to controls. This manuscript adds a new understanding to the gene-behavior relationship in the context of ASD-related phenotypes. However, there are several flaws to be tackled.

1. Terms like "main effect of group" and "marginal effect of group" are included in the content, but it would be better to further clarify them since readers are unfamiliar with these concepts. 

2. Although the manuscript focuses on women carrying the FMR1 premutation, it does not fully consider potential gender differences in the manifestation of prosodic traits. Females with the FMR1 premutation generally exhibit a milder phenotype, and this could affect the prosodic features observed. Explain this choice in detail.

3. A more detailed discussion of how these subclinical traits may interact with cognitive functioning could be included

4. It seems that the study does not account for potential environmental influences on speech prosody, such as socioeconomic status, educational background, or cultural differences in speech patterns. Or they just not needed in this manuscript?

5. The study employs acoustic measures of intonation and rhythm (e.g., F0 range, syllabic oscillation), it seems to not consider functional communication aspects or social pragmatic skills directly.